# Antimicrobial Use in Broilers Reared at Different Stocking Densities: A Retrospective Study

**DOI:** 10.3390/ani10101751

**Published:** 2020-09-26

**Authors:** Jacopo Tarakdjian, Katia Capello, Dario Pasqualin, Giovanni Cunial, Monica Lorenzetto, Luigi Gavazzi, Grazia Manca, Guido Di Martino

**Affiliations:** 1Istituto Zooprofilattico Sperimentale delle Venezie, Viale dell’Università 10, 35020 Legnaro (PD), Italy; jtarakdjian@izsvenezie.it (J.T.); kcapello@izsvenezie.it (K.C.); dpasqualin@izsvenezie.it (D.P.); gcunial@izsvenezie.it (G.C.); mlorenzetto@izsvenezie.it (M.L.); gmanca@izsvenezie.it (G.M.); 2Veterinary Practice of Poultry Sector, 25100 Brescia, Italy; luigi.gavazzi@amadori.it

**Keywords:** broiler, DDD, antimicrobial use, stocking density, welfare

## Abstract

**Simple Summary:**

Broiler farms are authorized to house animals at different stocking densities (kg live weight/m^2^) provided that ammonia, carbon dioxide, temperature, and humidity levels are kept under specific limits set by European Union legislation. The role of stocking density on poultry health and welfare, and consequently on antimicrobial use, is still under debate. Herewith, we analyze veterinary prescription data for 2017 pertaining to a representative sample of Italian broiler farms that were authorized to house animals at 33 or 39 kg/m^2^. The prescribed amounts of antimicrobials in each farm were converted into a total of Defined Daily Doses per year, a non-dimensional unit proposed by the European Medicines Agency (EMA). No differences in terms of antimicrobial use, mortality, and feed conversion rates were noted between the two considered groups, suggesting a minor role of stocking density on animal health and welfare.

**Abstract:**

According to the Directive 2007/43/EC, broiler farms can house animals up to 39 kg/m^2^, provided that specific environmental requirements are met. However, limited information is available about the effects of stocking density (SD) on broiler health and welfare, including the need for antimicrobial use. In this study, annual data on mortality, feed conversion rate, and antimicrobial use (AMU) are compared between broiler farms with stocking densities of 39 kg/m^2^ (N = 257) and 33 kg/m^2^ (N = 87). These farms were distributed throughout Italy and belonged to the same integrated poultry company. Antimicrobial use data were obtained from each farm and production cycle; AMU was expressed using the defined daily doses (DDD) method proposed by EMA. The annual AMU per farm was calculated as the median AMU over all cycles. Stratified analysis by sex and geographical area (Italy vs Northern Italy) showed no significant effect of stocking density on broiler mortality, feed conversion rate, and AMU. However, a higher AMU variability among farms with 39 kg/m^2^ stocking density vs. those with 33 kg/m^2^ was found. This study indicates that AMU does not apparently vary between animals reared at different stocking densities in intensive farms.

## 1. Introduction

Chicken meat consumption has increased considerably during the last decades, and this upward trend continues worldwide [1]. The poultry production sector owes its success mainly to the characteristics of chicken meat itself, as this meat is a valuable source of proteins and contains low levels of saturated fats. In addition, poultry meat consumption is not prohibited by any religion and can usually be purchased at convenient market prices [2]. To meet the ever-growing market demand of chicken meat, broiler production has undergone a substantial ‘acceleration’ in terms of the growth rate and density of reared birds, mainly by means of intense genetic selection programs [3] and high stocking density rearing systems [4]. The outcome of these processes led to increased public awareness towards animal health and welfare, particularly the consequences of genetic selection (e.g., cardiovascular and musculoskeletal disorders) as well as those associated with high SD due to excessive litter moisture (e.g., footpad and thigh lesions) [5,6]. Adequate ventilation, nutrition, and bedding materials are key to successfully improve broiler performance and minimize losses [5,7]. Yet, conflicting results have been reported on the possible effect of SD on broiler health and welfare [6,8]. Indeed, high SD has been reported to impact negatively on body weight gain (BWG) and feed intake (FI) because of the limited access to feeders and drinkers, as well as increased competitive behaviors [4,9]. These drawbacks of high SD can be mitigated, to some extent, by adapting the design and number of feeding and drinking units in the poultry house so as to allow all birds to have access to them in a homogeneous way. Moreover, broiler welfare seems to be influenced more by the housing conditions as a whole rather than by the SD per se [6], and these observations have set the basis for the current European Union (EU) legislation for the protection of chickens kept for meat production. Indeed, according to EU Council Directive 2007/43/EC [10], broiler farms are allowed to increase the maximum SD from 33 to 39 kg/m^2^, provided that specific environmental requirements are met. For instance, each holding should be designed in such a way that the concentration of ammonia never exceeds 20 ppm and the concentration of carbon dioxide never exceeds 3000 ppm. Moreover, indoor temperature should not exceed the outdoor temperature by more than 3 °C when this latter exceeds 30 °C in the shade, and the average relative humidity measured during 48 h should not exceed 70% when the outdoor temperature is below 10 °C. As high ventilation rates positively influence nitrogen, moisture, and heat dissipation, the incidence of skin lesions is also reduced [11]. This explains why, in the case of high SD, feedback at the slaughterhouse is requested by Directive 2007/43/EC [10], including reports on mortality (daily and cumulative) and incidence of footpad dermatitis. An excess of mortality linked to a higher SD may lead to the revocation of the authorization for the farmer. Higher levels of antimicrobial use (AMU) can be hypothesized to be necessary to maintain mortality low in farms with high SD. To date, the quantification of AMU is a subject of great interest, as it allows for the monitoring of antimicrobial consumption in the veterinary sector, which has often been seen as the main driver of antimicrobial resistance (AMR) [12]. To raise awareness and promote a responsible use of antimicrobial compounds, the European Medicines Authority (EMA) has developed a harmonized dose-based approach for collecting and processing AMU-related data [13]. On these grounds, we compare AMU data, mortality, and feed conversion rates of several Italian broiler farms rearing animals at different stocking densities.

## 2. Materials and Methods

### 2.1. Data Collection

Annual data on mortality, feed conversion, body weight produced, number of slaughtered animals, and AMU were collected in 2017 from 344 broiler farms (Figure 1), 257 of which were authorized to house animals at 39 kg/m^2^ (D39), whereas 87 farms could not exceed 33 kg/m^2^ (D33). Production in these farms accounted for approximately 18% of Italy’s total broiler production [14]. All farms belonged to one of Italy’s main poultry companies. Overall, 1028 grow-out cycles were investigated at D39 and 348 at D33. Cycles lasted 35–54 days.

The poultry company is integrated and therefore controls all production phases, including husbandry practices, genetics of the birds, feeding and vaccination programs. Therefore, variability among farms is expected to be minimized. The quantification of AMU followed the dose-based approach proposed by the EMA [15]. Raw data on AMU were collected by farm veterinarians for each production cycle and were converted into Italian Defined Daily Doses (DDDita) and expressed as DDDita/kg as proposed by Caucci et al. [16], thereby allowing for a high level of accuracy since DDDita/kg also takes into account the potency of the marketed products sold in Italy. To compute the number of administered doses (nDDDita), the total mg of administered active principles per cycle was divided by the maximum daily dosage given in the Summary of Product Characteristics (SPC), since veterinarians usually report to administer the highest posology. nDDDita is further divided by the total kg of live weight at slaughter according to the following formula:(1)$$\mathrm{DDDita}/\mathrm{kg}=\frac{\mathrm{tot}\;\mathrm{mg}\;\mathrm{of}\;\mathrm{Active}\;\mathrm{Substance}}{\mathrm{Defined}\;\mathrm{Daily}\;\mathrm{Dosage}}*\frac{1}{\mathrm{tot}\;\mathrm{kg}\;\mathrm{of}\;\mathrm{live}\;\mathrm{weight}}.$$

### 2.2. Statistical Analysis

In order to assess the data distribution for AMU, feed conversion, and mortality rates, descriptive statistics were computed. Because departure from normality was observed for all parameters, non-parametric tests were applied to test the statistical significance of differences in AMU, feed conversion, and mortality rates between D33 and D39 farms. Specifically, the Mann-Whitney two-sample statistic was applied after having checked the equality of variances by means of the Levene’s robust test statistic. Additionally, the aforementioned tests were used for the stratified analyses by sex and geographical area (national level and Northern Italy). Statistical significance was set to *p* < 0.05. Data analysis was performed using software STATA v.12.1 (Statacorp LP Statistics, College Station, TX 77845, USA).

## 3. Results

Descriptive statistics, expressed using quartiles of AMU, feed conversion, and mortality rates, are reported in Table 1 (national level). There were no significant differences in these variables between D33 and D39 farms, neither when stratifying by sex and geographical area.

## 4. Discussion

In this study, we investigated whether 33 and 39 kg/m^2^ stocking densities could be associated with different values of AMU, feed conversion ratio, and mortality in broiler farms, but no significant differences were found. Previous studies suggested that SD is not a major driver for broiler health and welfare [6]. This formed the basis for the current EU legislation allowing farms to increase SD, provided that they assure proper microclimate conditions and post-mortem examination at the slaughterhouse does not reveal any problem in terms of excess mortality and footpad dermatitis. A potential limitation of this system is that a harmonized lesion scoring system for contact dermatitis, parasitic, and systemic diseases has not yet been formalized at the EU level. Therefore, each EU Member State (MS) is tasked with the definition and application of appropriate and measurable criteria to assist the competent health authorities in the decision-making process. In Italy, health authorities perform documentation checks, including daily mortality rate (DMR) and cumulative daily mortality rate (CDMR), as well as dead on arrival (DOA) rate. The DOA should not exceed 1.5%, whereas CDMR should be lower than 2 + (0.12 * number of days). When these limits are exceeded, veterinary authorities may proceed with inspecting the rates of discarded animals and footpad lesion scores on at least 200 broiler feet, with limits being set at 2% and 100 points, respectively. Footpad lesions are evaluated on a 3-point scale, from 0 to 2, depending on the extent of the injury. Further non-compliance in terms of animal welfare should be communicated from the veterinary authorities to the poultry holding owner and to the competent authorities, which may consequently decide to decrease or not the SD [17]. However, this system is not without critical aspects either. For instance, concerning the cumulative daily mortality rate, farmers often claim that the hatchery conditions are the main cause for the 1st-week high mortality rates [18]. Authorities generally take those claims for granted and do not usually inspect hatcheries because EU law does not call for specific on-hatchery audits. Furthermore, footpad lesions are reported to be strongly related to litter moisture and season, rather than SD per se [19]. Finally, the threshold for DOA does not reflect average values (around 0.33%) [20] and is influenced by transport condition, rather than by the husbandry system itself [21]. No threshold for AMU is currently set by health authorities in relationship with SD, and it cannot be excluded that a higher AMU is needed to keep mortality and footpad lesions under control, as they are partly determined by enteric dysbiosis that worsen litter quality [22]. However, the present study seems to suggest that no higher AMU is needed at increased SD.

Data were obtained from a representative sample of broiler farms accounting for around 18% [14] of the national production. The fact that these farms belonged to the same poultry company guaranteed their comparability with respect to the many determinants of AMU, feed conversion and mortality rates, such as different farm management practices, vaccination protocols, genetics, and so forth. On the other hand, our results should be confirmed in other farm management conditions as well. Furthermore, a possible limitation of the comparison made in this retrospective study is represented by the fact that the lower density farms have actually not been allowed to increase in density. It cannot be excluded that those reasons could also affect antibiotic intake.

The company participating in the present study declared to have implemented specific actions to reduce AMU, such as training courses for farmers regarding health management and animal welfare, improvements of biosecurity, ban on tetracycline and colistin use, interventions on drinking water quality, and high hygienic procedures at the hatcheries. Penicillins, sulfonamides, and amphenicols represented the most frequently administered antibiotic classes, accounting for 65.5%, 17.4% and 11.2% of the total AMU in 2017 in the farms under study. A significantly higher AMU was recorded in Northern Italy, which represents the most densely populated poultry area in the country. Densely populated poultry areas are considered to be more susceptible to infectious diseases, due to the close proximity of holdings and to the strict integration of the poultry industry (connections of personnel, feeding, animals, etc.) [23].

No differences in feed conversion and mortality rates were observed, which implies that SD might play a generally minor role on health and welfare of intensively reared poultry. This is in agreement with Dawkins et al. [6]. However, Hall [9] found higher mortality rates in flocks with 40 vs. 34 kg/m^2^ SD during the 1st and the 4th week of production, but the explanation for this difference did not clearly point to SD alone. Abubados and colleagues [4] investigated 28, 37, and 40 kg/m^2^ SD effects on broiler feed conversion rates without observing any decrease, which is in line with our findings.

## 5. Conclusions

This retrospective study conducted in a major Italian poultry company revealed that broiler farms rearing animals at 33 and 39 kg/m^2^ SD did not exhibit differences in terms of AMU, mortality, and feed conversion rates. If confirmed under different management conditions, these findings would further support the hypothesis that a higher SD does not necessarily affect broiler health and welfare, nor represent a risk factor for AMR by increasing the need for AMU.

## Figures and Tables

**Figure 1 animals-10-01751-f001:**
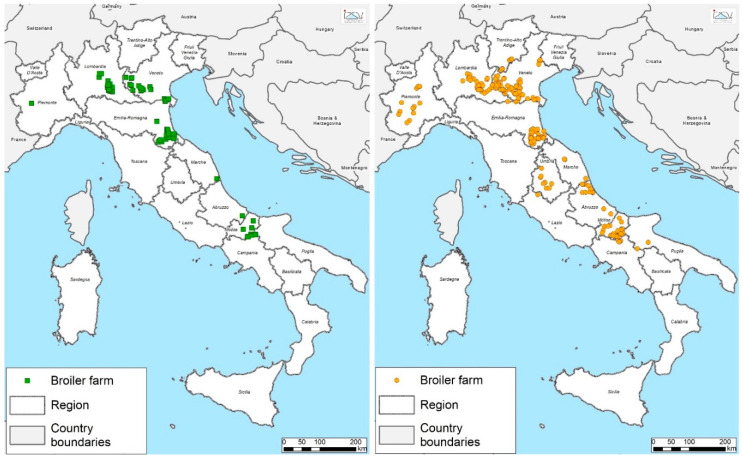
Geographical distribution of the 344 broiler farms enrolled in the study. Green squares indicate farms with the maximum stocking density of 33 kg/m^2^, and yellow circles indicate farms with a stocking density of 39 kg/m^2^.

**Table 1 animals-10-01751-t001:** Descriptive statistics of antimicrobial use, feed conversion rate, and mortality in 344 broiler farms in Italy.

Variables	Stocking Density (kg/m^2^)	N	Q1	Median	Q3
	overall (N = 344)				
Antimicrobial use (DDDita/kg)	33	87	0.16	0.44	1
	39	257	0.13	0.47	1.39
Feed conversion ratio	33	87	1.68	1.72	1.75
	39	257	1.68	1.71	1.74
Mortality (%)	33	87	3.5	4.1	5
	39	257	3.6	4.3	5.2
	sex: Female (N = 46)				
Antimicrobial use (DDDita/kg)	33	18	0	0	0.53
	39	28	0	0	0.41
Feed conversion ratio	33	18	1.63	1.69	1.74
	39	28	1.68	1.72	1.77
Mortality (%)	33	18	2.9	3.45	4
	39	28	2.75	3.45	3.8
	sex: Male (N = 130)				
Antimicrobial use (DDDita/kg)	33	32	0.17	0.25	0.69
	39	98	0.14	0.29	1.21
Feed conversion ratio	33	32	1.68	1.71	1.74
	39	98	1.68	1.71	1.73
Mortality (%)	33	32	3.75	4.5	5.75
	39	98	3.9	4.9	6

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
