# Peer review of "Antimicrobial Use in Broilers Reared at Different Stocking Densities: A Retrospective Study"

_animals, 2020, doi:10.3390/ani10101751_

Round 1

Reviewer 1 Report

Some minor corrections to the MS were made using Text Edits in Adobe PDF at the following line numbers:

26, 71, 72, 74, 75, 83, 195 and 2015  (See text for changes)

Also, in Table 1 for DDDita/kg -39  the value for the median is 0.,25.  Remove the comma after the period.  Should read 0.25.

Interesting report. 

Author Response

Some minor corrections to the MS were made using Text Edits in Adobe PDF at the following line numbers:

26, 71, 72, 74, 75, 83, 195 and 2015 (See text for changes)

Also, in Table 1 for DDDita/kg -39 the value for the median is 0.,25. Remove the comma after the period. Should read 0.25.

Interesting report.

Dear reviewer,
attached you can find the manuscript revised accordingly with your comments. You can check the amendments by using the "track changes" function in Microsoft Word.

Thank you for your revision and your comment.
Kind regards

Jacopo Tarakdjian

Reviewer 2 Report

The submitted work was well-designed with sufficient sampling and analyses were performed appropriately. The conclusion was supported by the data. Just one minor question, what’s the point for showing the 232 farms data independently as Table 1? This data technically were already included in Table 2, right?  

Author Response

The submitted work was well-designed with sufficient sampling and analyses were performed appropriately. The conclusion was supported by the data. Just one minor question, what’s the point for showing the 232 farms data independently as Table 1? This data technically were already included in Table 2, right?  

Dear reviewer, thanks for your revision.
You are right, Table 1 data are actually included in table 2. We aimed to provide a comparison between national and local data. Anyway we agree with your observation. We accordingly removed table 1 and kept table 2 only, since the two data sets did not significantly differ.

Kind regards,

Jacopo Tarakdjian
